# Gastric Cancer: Advances in Carcinogenesis Research and New Therapeutic Strategies

**DOI:** 10.3390/ijms22073418

**Published:** 2021-03-26

**Authors:** Lornella Seeneevassen, Emilie Bessède, Francis Mégraud, Philippe Lehours, Pierre Dubus, Christine Varon

**Affiliations:** 1Biological and Medical Sciences Department, University of Bordeaux, INSERM, BaRITOn, U1053, F-33000 Bordeaux, France; lornella.seeneevassen@u-bordeaux.fr (L.S.); emilie.bessede@u-bordeaux.fr (E.B.); francis.megraud@chu-bordeaux.fr (F.M.); philippe.lehours@chu-bordeaux.fr (P.L.); pierre.dubus@u-bordeaux.fr (P.D.); 2Centre National de Référence des Helicobacters et Campylobacters, Centre Hospitalier Universitaire de Bordeaux, F-33000 Bordeaux, France; 3Department of Histology and Pathology, Centre Hospitalier Universitaire de Bordeaux, F-33000 Bordeaux, France

**Keywords:** stomach cancer, *Helicobacter pylori*, cancer stem cell, microbiota, EMT, microenvironment, biomarkers

## Abstract

Gastric cancer’s bad incidence, prognosis, cellular and molecular heterogeneity amongst others make this disease a major health issue worldwide. Understanding this affliction is a priority for proper patients’ management and for the development of efficient therapeutical strategies. This review gives an overview of major scientific advances, made during the past 5-years, to improve the comprehension of gastric adenocarcinoma. A focus was made on the different actors of gastric carcinogenesis, including, *Helicobacter pylori* cancer stem cells, tumour microenvironment and microbiota. New and recent potential biomarkers were assessed as well as emerging therapeutical strategies involving cancer stem cells targeting as well as immunotherapy. Finally, recent experimental models to study this highly complex disease were discussed, highlighting the importance of gastric cancer understanding in the hard-fought struggle against cancer relapse, metastasis and bad prognosis.

## 1. Histological and Molecular Classifications

Gastric cancer is the most common stomach malignancy among lymphomas, sarcomas, gastrointestinal stromal and neuroendocrine tumours. This heterogeneous disease has different phenotypes and most cases are gastric adenocarcinomas (GC) [1]. The two main GC, according to topographic sites, are the cardia and the non-cardia GC. The complexity of GC explains the diverse histological classification systems which exist [2,3].

The most commonly used classification is the Laurén classification which divides GC into intestinal, diffuse and intermediate subtypes, according to their histological phenotypes. Intestinal type GC, characterised by malignant cohesive epithelial cells and intestinal-type glandular differentiation infiltrating the tissue, is the most common type occurring in about 54% cases while diffuse type GC, found in about 32% cases, contains poorly differentiated and poorly cohesive tumour cells [2]. GC can also be classified since 2010, according to the WHO classification, into tubular, papillary, mucinous, poorly cohesive (including signet ring cell carcinoma) and mixed carcinomas. While the poorly cohesive including signet ring cell carcinomas corresponds to the diffuse subtype of the Laurén classification, the tubular, papillary and mucinous types correspond to the intestinal subtype [2]. The Carneiro system distinguishes 4 categories of gastric tumours: glandular, isolated cell, solid and mixed, based on their immunophenotype such that it further divides the Laurén intestinal subtype into tumours with intestinal, gastric or mixed differentiation, according to the expression of specific markers (MUC6, MUC5AC mucins and TFF1 peptide gastric markers or MUC2, CDX2, CD10 and pepsinogen-1 intestinal markers, amongst others) [3]. Finally, the Goseki classification also divides GC into 4 groups according to intracellular mucin production (poor or rich) and level of tubular differentiation (poorly or well differentiated) [2].

GC vary not only histologically, but also molecularly. The most commonly mutated gene in GC (40% cases) is the tumour suppressor gene *TP53*, key regulator of cell genomic stability [4]. Diffuse type GC are often characterised by somatic or hereditary mutations of *CDH1* gene, coding for E-cadherin, a cell junction protein whose invalidation participates to the “independent cell” GC phenotype. Epigenetic alterations such as gene inactivation-associated hyper-methylations in CpG islands can be observed in certain cancer-related genes (*APC*, *K-RAS*, *hMLH1*, *CDKN2A*). This CpG island methylator phenotype (CIMP) is an early event in GC and can be found in contiguous normal tissue associated with *H. pylori* infection [2,5,6].

Based on these molecular differences, the Cancer Genome Atlas Research Network (TCGA) has proposed a molecular classification dividing GC into 4 groups: Epstein Barr Virus (EBV)-positive (EBV), microsatellite instable (MSI), genomically stable (GS) and chromosomal instability (CIN) [7]. Array-based somatic copy analysis, whole-exome sequencing, array-based DNA methylation profiling, mRNA and miRNA sequencing, reverse-phase protein assay, microsatellite instability testing and whole-genome sequencing, used on 295 tumour samples in comparison with the germline profile, showed 8.8% EBV, 21.7% MSI, 19.7% GS and 49.8% CIN cases. The last subtype is mainly associated to the intestinal-type histology [7]. The Asian Cancer Research Group (ACRG) also proposed a 4-group classification based on mRNA expression, somatic copy number and targeted gene sequencing: microsatellite instability (MSI), microsatellite stable and epithelial to mesenchymal transition phenotype (MSS/EMT), microsatellite stable and presence of TP53 (MSS/TP53+) or no TP53 signature (MSS/TP53-) [8]. These two molecular classifications use advanced molecular techniques which are not feasible in practice for individual therapy decisions. To overcome this challenge, Setia et al. proposed techniques available in routine diagnostic practice like in situ hybridisation and immunohistochemistry and identified 5 GC groups: EBV-positive (5% cases), mismatch repair-deficient (16%), aberrant E-cadherin expression (21%), aberrant p53 expression (51%) and normal p53 expression (7%) [9]. Furthermore, Li et al. used the TCGA RNA-sequencing data to identify differentially expressed set of genes in diverse tumour types for use as diagnosis biomarkers and drug development [10].

Finally, recent developments in single cell methods allows more precise deciphering of GC cells heterogeneity, though not applicable as routine diagnosis. Zhang et al. propose a GC transcriptome atlas after analysis of 22,677 cells from 3 non-tumorous samples and 9 tumours [11]. Comparison of malignant epithelium to no malignant ones put into evidence overexpression of tumour specific genes like *REG4*, *TFF3*, *CLDN4* and *CLDN7*. Genes involved in gastric mucus and enzyme secretion like *GKN1*, *PGC*, *MUC5AC* and *LIPF* were more expressed in non-tumorous epithelium [11]. In order to compare the cellular genetic profiles obtained to known histopathological classifications, tumour samples with known histology were added to analysis. Analysed malignant cells clustered 5 main cell groups: C1 having 96.9% cells from diffuse-type sample, C2 composed of 97.1% cells from intestinal-type sample, C3 with mixed-type and intestinal- type cells, C4 containing cells mainly from one of the intestinal-type samples and C5 with cells from EBV+ patients. Interestingly, C4/C5 seem to be correspond to novel subtypes, molecularly different from C1, C2 and C3 which correspond to the Lauren classification subtypes [11].

These GC classifications (Figure 1) are the key to the comprehension of this disease and its underlying mechanisms, thus opening new pathways for novel targeted therapies since different GC types can be related to different risk factors, prognosis, and treatment management.

## 2. Epidemiology and Risk Factors

Gastric Cancer is the 5th most common and the 3rd deadliest cancer worldwide, according to the latest report of the International Agency for research on Cancer [12]. Its geographic incidence remains heterogenous with most cases occurring in Eastern Asia (619,226 in 2018), and men being twice as much affected as women. Interestingly, a decrease in GC incidence has been noted in Western Europe and North America over the past decade [13].

Different risk factors are known to affect gastric cancer incidence, among which *Helicobacter pylori* (*H. pylori*) infection is the essential one. *H. pylori* infection involves almost 50% of the world’s population and induces chronic inflammation of the gastric mucosa, 5 to 15% of which evolve in gastric and duodenal ulcers and less than 1% in GC. The Correa cascade shows how intestinal type GC begins with *H. pylori*-induced chronic gastritis followed by gastric atrophy, intestinal metaplasia, and dysplasia [14]. Intestinal GC subtype is mainly caused by environmental factors while diffuse GC subtype also includes hereditary and genetic factors. About 10% intestinal GC cases are associated to infection with EBV and interestingly, EBV-positive GC are mostly associated to MSI molecular type and are related to less worst survival prognosis than diffuse/mixed GC subtypes [15,16]. 

The overall decrease in GC incidence can mostly be attributed to the decrease in intestinal type GC due to a lower *H. pylori* prevalence and modified dietary habits [13,17]. The better comprehension of intestinal type GC over the past decade has largely contributed to this, and the Correa cascade has provided a window for its prevention and early detection. Studies show that *H. pylori* eradication treatments, both as primary and as secondary prevention, reduces the risk of GC development. It also decreases the rate of metachronous GC in patients having early GC and improves the baseline of gastric corpus atrophy grading [18,19].

Conversely, diffuse GC subtype, not only related to *H. pylori* infection, is not yet well understood, and still has an increasing incidence. Due to its predominant genetic and hereditary origin, diffuse type GC seems to be more frequently described in young patients (<45 years-old) and to correlate with worse disease-free survival while intestinal type GC with chronic gastric inflammation seems to appear mostly after 45 years-old (around 70 years-old) [20]. GC trends not only show a distinct evolution ratio between the two GC subtypes but also between GC localisation, with an increase in cardia GC incidence compared to distal GC which is mostly related to *H. pylori* infection [13].

Despite the high GC incidence, GC survival rates seem to be better in Eastern Asia with a 64.2% 5-year survival [12] compared to North America (2.7%) or Europe (8.5%). Indeed, risk factors comprehension has helped develop strategies against GC and in Eastern Asia screening programs have been established allowing early detection of GC. More than 50% of cases are diagnosed at an early stage in Japan for example, against 27% in the USA [13]. Asian GC cases are mostly non-cardia which, again due to *H. pylori* eradication campaigns and programs, can be detected early. Also, secondary preventive strategies like endoscopic screening and high-risk histology surveillance has been successful and will further contribute to decrease *H. pylori*-induced GC [18]. In North America and Europe, non-cardia GC incidence has decreased while cardia GC rates remain stable and is possibly increasing [13]. In addition, non-cardia GC seems to decrease in older populations (>50 years-old) and increase in younger persons, especially women, which could be linked more to autoimmune gastritis and less to *H. pylori*-related gastritis [15]. *H. pylori* eradication does not affect the cardia GC trend [18] and it is important to find precursor lesions or biomarkers allowing their early detection and maybe treatment. 

Still, GC incidence does not follow *H. pylori* prevalence, showing the role of other risk factors like the host’s genetics, *H. pylori* strain characteristics as well as environmental factors like salty diet in this disease. Indeed, Eastern Asian populations harbour highly virulent *H. pylori* strains possessing variants of bacterial pathogenicity factors associated with GC such as VacA, CagA oncoprotein and other proteins including BabA and SabA outer membrane proteins [21,22]. Chronic inflammation plays an important role in gastric carcinogenesis. A recent European Prospective Investigation into Cancer and Nutrition (EPIC) study focussed on the association between inflammatory potential of diets and the risk of cardia and non-cardia GC: 476,160 subjects from 10 European countries were followed for 14 years and the results showed that the inflammatory potential of diets is associated with an increased risk of GC. Low-grade chronic inflammation induced by food intake may be associated with cardia GC risk, the pattern being less consistent for non-cardia GC [23].

Furthermore, many cancers including GC and colorectal cancers have been linked to the consumption of red and/or processed meat [24]. These contain either carcinogens like heterocyclic aromatic amines and polycyclic aromatic hydrocarbons, produced by cooking at high temperature, or N-nitroso-compounds and polycyclic aromatic hydrocarbons in cured or smoked meat. In addition, processed meat contains much salt, one of the risk factors for GC, along with tobacco smoking and low vegetable and fruit intake which are sources of antioxidant vitamins [25].

Apart from external elements, intrinsic factors such as host susceptibility also affect GC. Almost 40% hereditary diffuse GC are due to the invalidating mutation of *CDH1* [26]. Moreover, whole-exome sequencing analyses show other germline mutations than *CDH1*, for example in tumour suppressor genes *SDHB*, *CTNNA1*, *STK11* and in genes involved in DNA repair *PALB2*, *BRCA2* and *ATM*. In addition, patients suffering from syndromes affecting DNA repair genes, *TP53*, *APC* as well as tumour suppressor genes (*BRCA*) are more likely to develop GC [2,5,6].

## 3. Gastric Carcinogenesis

### 3.1. A Helicobacter Disease

*Helicobacter pylori*, the major risk factor of GC, colonises the gastric mucosa and induces chronic gastric inflammation. Nevertheless, most infected individuals do not develop GC. Possible reasons might be the heterogeneity of the bacterial genome and its different virulence factors. The *cag* pathogenicity island (*cag* PAI) encodes the major virulent factors of the bacterium, which are CagA oncoprotein and proteins forming a type IV secretion system, normally related to the bacterial conjugation system allowing it to exchange DNA with other bacteria. Other pathogenic factors associated to GC include adhesins (BabA, SabA), outer membrane proteins (OipA, HomB), and VacA cytotoxin which is encoded by a *cag*-PAI independent locus [21]. Many studies have described the pro-oncogenic role of these different factors, either by being translocated into the host cells, by translocating other factors into host cells, or by facilitating gastric mucosa colonisation by *H. pylori*. CagA, the first described bacterial oncoprotein, is injected by the type IV secretion system into the cytoplasm of host gastric epithelial cells where it interacts with different signalling pathways. CagA destabilises cellular junctions and apico-basal cell polarisation, activates pro-inflammatory and oncogenic signalisation pathways leading to disturbance of the gastric epithelium integrity, differentiation and self-renewal [21]. Still, malignant transformation mechanisms are not fully known. 

Horvat et al. showed that *H. pylori* inhibits p14ARF tumour suppressor in a CagA dependent manner by inducing its degradation [27]. p14ARF is a critical tumour suppressor having important functions in oncogenic stress regulation. The chromosome region on which is located this gene is often deleted in primary GC and hypermethylation inactivates its expression in more than 30% of GC cases. The use of isogenic *H. pylori* mutants showed that CagA is responsible for the overexpression of E3 ubiquitin ligase, TRIP12, inducing ubiquitination and degradation of p14ARF protein in *H. pylori* infected cells and inhibition of autophagy in a p53-independent manner. TRIP12 is also overexpressed in the gastric mucosa of *H. pylori*-infected patients [27]. *H. pylori* also promotes gastric carcinogenesis in a p53-dependent manner. Costa et al. showed that this bacterium can reduce the expression of transcription factor USF1, known for its p53 stabilizing role in response to genotoxic stress [28]. The consequent p53 proteasomal degradation participates to gastric carcinogenesis promotion. In addition, *H. pylori* delocalises to foci nearby cell membranes and prevents USF1 /p53 nuclear translocation, altering their transcriptional function, one of which is the protection of gastric cells against *H. pylori* infection [28]. 

*H. pylori* infection also interferes with other tumour suppression pathways. This bacterium can downregulate the tumour suppressor Receptor for Activated C Kinase 1 (RACK1) which is physiologically involved in the modulation of NF-κB signalling pathway activity. By doing so, *H. pylori* promotes the upregulation of integrin β-1, leading to the upregulation of the NF-κB signalling pathway involved in *H. pylori*-induced carcinogenesis. In addition, this study shows that RACK1 is downregulated in GC tissue compared to normal tissue distant to tumour and is correlated to poor prognosis of patients [29].

*Helicobacter pylori* can use the host mechanisms to increase its virulence. It can control the activation of c-Abl kinase to maintain CagA virulence factor phosphorylation [30]. In addition, it has been recently shown that *H. pylori* not only induces c-Abl kinase activity but also alters the localisation of the activated protein. c-Abl becomes cytoplasmic, promotes cell migration and prevents apoptosis thus participating to *H. pylori*-related gastric carcinogenesis [31].

Gastric mucosal barrier disruption by *H. pylori* infection and inflammation also plays a role in gastric carcinogenesis [21,32]. The gastric mucosa contains epithelial cells with transmembrane and peripheral scaffolding proteins among which Claudins with a role in the regulation of tight junctions’ permeability [32]. Hagen et al. showed that Claudin-18 is decreased in models of GC infected with *H. pylori* as it is in GC patients where this loss is associated with an aggressive phenotype and poor prognosis [33]. Targeted deletion of *CLDN18* gene encoding Claudin-18 results in pre-neoplastic stomach lesions in 7-weeks mice. Interestingly, 20 weeks post-infection, *H. pylori* is not able to colonise Claudin-18 deficient mice but, atrophy of the gastric corpus as well as high level of dysplasia and gastrointestinal neoplasia resembling human intramucosal carcinoma are noted in sham and *H. pylori*-infected mice [33]. 

Cell adhesion molecules modifications induce Epithelial-to-Mesenchymal Transition (EMT) where epithelial cells transdifferentiate and transit from their epithelial phenotype to a mesenchymal phenotype and in so doing, migrate or invade (in case of tumours). *H. pylori* infection is known to initiate EMT in gastric epithelial cells [34,35]. Invalidation of *IQGAP1*, involved in adherent junctions stability and sequestration of β-catenin to the junctions, increases *H. pylori*-induced EMT in gastric epithelial cell lines in vitro and promotes *H. pylori*-induced pre-neoplastic lesions in vivo with 6 times more Gastric Intra-epithelial Neoplasia (GIN) in mice mutated for *IQGAP1* compared to wild-type mice [36]. A recent study shows that *H. pylori* also downregulates other adherence proteins like Afadin, having a role in tight junction stabilisation, and in so doing, induces EMT phenotype in GC cell lines. Afadin regulation is independent of CagA, type-IV secretion system and VacA virulence factors [37]. Also, *H. pylori* upregulates the expression and activity of various matrix metalloproteases family proteins (MMPs) in GC cells, exacerbating their invasive properties. The use of several signalling pathways inhibitors has revealed the role of Src, NFκB, ERK and JNK pathways in *H. pylori*-induced EMT and EGFR-modulated MMPs (MMP9, 3, 10) regulation [34,35,38]. 

CagA+ *H. pylori* infection dysregulates the Hippo signalling pathway, responsible for the control of stem cell properties and proliferation, in physiology. *H. pylori* induces Hippo oncogenic effector YAP1 overexpression both in vitro and in vivo in infected patients. Consequently, activation of YAP1/TEAD oncogenic pathway in gastric epithelial cells promotes EMT as well as the acquisition of intestinal metaplasia markers [39,40]. Moreover, the Hippo pathway seems to limit *H. pylori*-induced preneoplastic changes. The bacterial infection up-regulates oncoprotein YAP1 but also its negative regulator LATS2, a Hippo tumour suppressor kinase, in a coordinated biphasic manner, with an early temporary oncogenic YAP1 activation, followed by LATS2 activation leading to YAP1 phosphorylation and downregulation and thus, oncogenic signal restriction [40]. YAP1 paralogue TAZ is also activated after *H. pylori* infection, overexpressed in vivo, and is associated to EMT and to acquisition of tumorigenic and invasive properties in GC cells [41]. Furthermore, YAP1/TAZ seem to cooperate with β-catenin in Wnt signalling pathway to promote gastro-intestinal neoplasia [42].

Additionally, gastric carcinogenesis can occur after *H. pylori* infection through the regulation of connexins (Cxs). This bacterium can upregulate transcription factors like GATA-3 and PBX-1, having a role in the expression of Cx32, which is thus inhibited [43]. In addition, it can decrease histone acetylation levels which in turn regulates the expression of Cxs. Similarly, *H. pylori* modulates other connexins such as Cx43, Cx26 and Cx37, expressed in gastric tissue, either by promoter hypermethylation and expression inhibition, protein delocalisation or gene polymorphism induction [43]. 

*Helicobacter pylori* also seems to protect GC cells from anti-tumoral immune response. Programmed death (PD) and programmed death ligand 1 (PD-L1) expression is one of the immune tolerance mechanisms preventing T-cells from attacking one’s own tissues. PD-L1 expression was detected on tumour cells showing that tumour cells seem to exploit this immune-checkpoint pathway to escape cytotoxic T-cells-induced programmed cell death [2]. *H. pylori* is able to stimulate PD-L1 expression in gastric epithelial cells. It was recently found that Shh pathway is involved in this CagA dependent process. PD-L1 expression increases in *H. pylori*-infected organoid cultures and Shh pathway inhibitor GANT61 is able to counteract this response [44]. In addition, anti-PD-L1 treatment of gastric organoids co-cultured with autologous patient cytotoxic T lymphocytes and dendritic cells provoked organoids death. The use of this PD-L1 mechanism causes epithelial cell survival and protection against immune response, leading to GC progression and interestingly, *H. pylori* is not the only pathogen using this mechanism. A recent study showed that among EBV-positive tumours, those with high viral load was correlated to higher tumour cells PD-L1 expression and worse patient prognosis compared to those with lower EBV viral load [45].

*H. pylori* infection induces the NF-κB pathway which increases Peroxiredoxin 2 (PRDX2) expression in GC [46]. This antioxidant enzyme plays an important role in the protection of cells from oxidative stress by scavenging H2O2 and ROS from cells and seems to be used by cancer cells as a defence mechanism. PRDX2 expression is high in GC tissues and correlates with low survival of patients [46].

*H. pylori* not only alters the hosts’ mechanisms to induce disease, but also undergoes genetic modifications in vitro and in vivo when in contact with the carcinogenic environment. Whole genome sequencing revealed a total of 180 unique single nucleotide polymorphisms (SNPs) in differently virulent *H. pylori* strains and in strains harvested in low-iron and high-salt carcinogenic conditions compared to reference *H. pylori* genome [47]. Common SNPs, including one within *fur* gene (FurR88H) encoding a ferric uptake regulator, were found in strains cultivated in cancer-like environment. This FurR88H variant seemed to appear after only 5 days exposure to the carcinogenic environment and *fur* sequencing in clinical isolates showed that 17% of strains coming from patients with premalignant lesions had the FurR88H variant compared to 6% strains from non-atrophic gastritis patients [47]. Furthermore, a genome-wide association study (GWAS) on 173 *H. pylori* isolates from European population revealed SNPs and genes, *babA* and *cag* pathogenicity island genes, as well as non-synonymous changes in several less well-studied genes, that differed in GC patients compared to those suffering from gastritis. The authors conclude that these bacterial factors differ enough in this pathology for a minor GWAS analysis to detect them [48]. 

Nonetheless, host polymorphisms can also render them more sensitive to GC development in presence of *H. pylori*, for example if there is less production of anti-inflammatory cytokines, activation of pro-inflammatory cyclooxygenases and oxidative damage [21]. 

*Helicobacter pylori* infection indeed affects GC status through a large variety of ways (Figure 2) and understanding the mechanisms will open paths to more targeted and efficient therapies. 

### 3.2. A Stem Cell Disease

Despite the major influence of environmental factors, GC remains a heterogenous and multifactorial disease. Tumour cells are different from one another in terms of the mutations they carry, their sensitivity to drugs, their phenotype as well as their tumorigenicity. Indeed, ever since the identification of Cancer Stem Cells (CSCs), cancer cells having stem-like properties, in acute myeloid leukaemia in 1995, several studies have demonstrated the presence of such cells in solid tumours like breast, brain, colon [49,50,51,52] as well as gastric tumours [53,54,55]. GC are diverse entities not only due to the different histological phenotypes between patients but also due to the presence of different cell populations within one tumour mass, among which Gastric Cancer Stem Cells (GCSCs) form a very small proportion, ranging from 0,1 to 3,5% according to Nguyen et al. [55]. These cells have self-renewal as well as asymmetrical division capabilities allowing them to regenerate themselves and to generate new differentiated cells for proliferation and tumour mass enlargement. These highly tumorigenic cells are also known to resist to conventional chemo- and radiotherapies and to migrate and cause metastasis in patients [55,56]. Studying them is crucial to decipher GC mechanisms and develop more specific and efficient therapies in GC.

Ever since their discovery, many methods have been developed taking advantage of their specific self-renewal, differentiation, and tumour initiation properties. In vitro sphere assays, in which cells are seeded at low density on non-adherent plates in the absence of serum and presence of growth factors, allow only GCSCs to survive and form spheres. In addition, only GCSCs have the capacity to generate new spheres after several passages. In vivo experiments are also used since CSCs can generate heterogenous tumours after subcutaneous xenograft in immunocompromised mice [57]. 

In addition, characterisation of GCSCs has allowed the discovery of markers which could be used to isolate them using Fluorescence-Activated Cell Sorting (FACS), for example. CD44, best known as being the hyaluronic acid receptor, is a marker of GCSCs as it is for many other solid tumours [50,55]. Nguyen et al. described CD44 and ALDH activity as being the most important GCSC signature*s* allowing the detection and the isolation of cells able to generate heterogeneous tumorspheres in vitro and tumours in vivo from non-cardia gastric carcinomas. The authors described ALDH+ cells as being part of the CD44+ cell population and as possessing drug efflux properties increasing their chemoresistance properties [55]. In addition, many other markers enriched in GCSCs like CD133, CD24, CD166 were described and can be used for GCSC study. These methods have allowed to shed a new light on GC as a stem cell disease and *H. pylori*, being the major GC risk and having a role in GCSC emergence, many studies associate GCSCs and *H. pylori*. 

*Helicobacter pylori* induces the emergence of cells with CSC phenotype and properties, through EMT, in infected patients’ gastric mucosa and in cell lines in vitro. Bessède et al. showed that only CD44+ cells seem to display mesenchymal phenotype compared to CD44– cells after *H. pylori* infection [35]. *H. pylori*-induced cell junction alterations also affect CSC properties. Deletion of IQGAP1 junction scaffolding protein increases neoplastic lesion in mice stomach and induces CD44 overexpression, mesenchymal phenotype, and CSC-like properties after *H. pylori* infection [36]. *H. pylori* infection also stimulates Lrig1+ gastric stem cell population, in a CagA-dependent manner, showing another way by which, this bacterium regulates gastric stemness phenotype. Lrig+ cells are enhanced in human GC tissues compared to normal mucosa [58]. CagA+ *H. pylori* infection induced-YAP1 and TAZ oncogenic pathway not only promotes EMT as discussed before, but also the acquisition of CSC phenotype as well as tumorigenic and invasive properties in vitro [39,40,41]. YAP1, TAZ and their target genes are significantly upregulated in GC and associated to poor prognosis [59]. Hippo pathway contribution to GCSC phenotype was demonstrated by an enrichment of YAP1 and its partners in CD44+ GC cells, and by the inhibition of GCSC tumorigenic properties in vivo after YAP1/TAZ/TEAD targeting by Verteporfin [59]. Moreover, activation of Hippo tumour suppressor kinase LATS2 by Leukaemia Inhibitory Factor (LIF), an interleukin-6 family cytokine, decreases GCSC tumorigenic properties and population of GC cell lines and patient-derived xenograft cells (PDX), again showing the implication of this pathway in GCSC phenotype [60]. High expression of LIF’s receptor (LIFR) in patients is associated to better survival rates and patients having high YAP1/TAZ and target genes expressions which is correlated to bad prognosis, survive better if they also have a high LIFR expression [60], showing a protecting role of LIF/LIFR signalling in this YAP1/TAZ-pro-GCSC context. 

CD44 plays a functional role in *H. pylori*-induced proliferation of gastric epithelial cells. It acts as a co-receptor of tyrosine kinase c-Met receptor and precipitates with phosphorylated c-Met and CagA. In this context, *H. pylori* fail*s* to induce epithelial proliferation in organoids derived from CD44-deficient mice stomachs showing the importance of CD44 in *H. pylori*-induced gastric carcinogenesis [61]. CD44+ CSCs presence at the invasive front of GC tumours indicate poor survival of patients compared to patients with no CD44 expression at invasive front and can be used as a prognosis marker [56].

CD44 exists in multiple isoforms containing at least 20 exons due to alternative mRNA splicing. CD44 variant 8-10 (CD44v8-10) has been identified as being predominantly expressed in gastric cancer cells and less in normal tissues. Exogenous expression of CD44v8-10 increase*s* the frequency of tumour initiation in immunocompromised mice. Finally, this CD44 variant, and not panCD44, can rescue the tumorigenic phenotype lost after panCD44 silencing [62].

Furthermore, another CD44 variant, CD44v9 is highly expressed in mouse GCSCs. Analysis of GC samples from 103 patients show a five-year survival rate lower in CD44v9-positive tumours compared to CD44v9-negative ones [63]. Capping actin protein of muscle Z-line α subunit 1 (CAPZA1) is a protein having an important function in actin polymerisation. Its overexpression in gastric epithelial cells inhibits the formation of autolysosome and results in the accumulation of CagA in host cells, increasing GC risk. In human GC cell line AGS, oxidative stress increases histone H3 acetylation of CAPZA1 promoter, increasing CAPZA1 expression, nuclear accumulation of CD44 transcription factor, β-catenin, and enhancing expression of epithelial splicing regulatory protein 1 responsible for the alternative splicing of CD44 into CD44v9 [64]. Moreover, CD44 variant 6 (CD44v6) does not seem to play a role in GC progression since CD44v6 overexpression does not affect GC cells growth rate, invasion, migration, or cell-cell aggregation. However, CD44v6 cells survive better than CD44v6- cells in presence of chemotherapy drug cisplatin suggesting a role of CD44v6 in GC chemoresistance [65].

A recent study analysed the role of Human Epidermal growth factor 2 (HER2), target of trastuzumab in the treatment of HER2+ gastric and breast cancers, on GCSCs and the underlying mechanisms. The authors demonstrated that HER2 overexpressing GC cells had increased stemness and invasiveness and were regulated by the Wnt/β-catenin signalling pathway [66].

The role of microRNAs (miRNAs) in the gastric carcinogenesis and in GCSCs control is also important. miR-181a_5p and miR-22-3p have been shown to be downregulated in GCSCs while miR-483-5p, miR-4270 and miR-16-5p are overexpressed. *A* recent study shows that miR-7-5p is also downregulated in GCSCs due to hypermethylation of its promoter region. miR-7-5p silencing increases GCSC invasion properties while its overexpression reduces spheroid formation and invasion by repressing Notch and Shh pathways in vitro and decreases tumour growth in vivo [67].

Due to their possible role in chemoresistance, relapse and metastasis, GCSCs comprehension (Figure 3) is a key step to understanding GC for biomarkers and therapy development. 

### 3.3. A Microenvironment Disease

The importance of the tumour microenvironment (TME) in tumorigenesis is being more and more explored. A tumour is a dynamic entity with constant communication*s* between the tumour cells and their rich surrounding environment, source of factors allowing and maintaining cancer cells phenotype and heterogeneity (Figure 4).

Most immune cells infiltrating tumours are Tumour Infiltrating Lymphocytes (TILs) consisting of CD3+ Tcells (CD8+ Tcytotoxic and CD4+ Thelper cells) as well as FOXP3+ Treg cells. The number of TILs present in tumours reflects the host immune response mechanism. Indeed, Yu et al. demonstrated that high CD8+ and CD3+ T cells infiltration in tumours are associated to better prognosis of patients compared to low infiltration [68]. Interestingly, CD8+ T cells seem to be less present in diffuse/mixed types GC compared to intestinal type and might be associated to worse outcome [69].

Tumour associated Neutrophils (TANs) are one of the most important stromal partners having a role in carcinogenesis. TANs are able to increase migration and invasive capacities of GC cells [70]. Increase in neutrophil blood count, in neutrophil/lymphocyte ratio as well as in TME infiltration with neutrophils reflect poor patient prognosis [71,72,73]. Li et al. demonstrate that TANs produce IL-17a which promotes EMT and GC progression by activating the JAK2/STAT3 pathway. IL-17a neutralising antibody was able to reverse TANs effect on GC progression [73]. 

Gastric cancer-associated fibroblasts (GCAFs), when activated, are able to enhance migration and invasion properties of gastric cells both in co-culture and in presence of GCAFs-conditioned medium. Activated GCAFs correlate with poor survival of patients and in vitro, paracrine action of factors from GCAFs make gastric cells resistant to 5-fluorouracil (5-FU), one of the conventional chemotherapies used in GC treatment [74]. A recent study shows the role of GCAFs in stemness, transformation and chemotherapy resistance. Low expression of Secreted Protein Acidic and Rich in Cysteine (SPARC) in GCAFs seems to be associated to low cell differentiation, low 5-year survival rate and 5-FU resistance [74]. Also, GCAFs produce large amounts of IL-6 which participate to the activation of the JAK/STAT signalling pathway. The use of anti-IL-6 neutralising antibody and inhibitor of JAK2/STAT3 decreases GCAFs effect on GC and GCAFs-induced peritoneal metastasis in vivo showing the role of TME and GCAFs in GC progression through the JAK/STAT pathway [75]. IL-6 crosstalk between tumour cells and GCAFs not only supports tumour growth, but also promotes CAFs activation through IL-6Rα. Loss of IL-6 and use of inhibitors of STAT3 and MEK/ERK pathways suppress tumorigenesis in 3D organotypic and tumoroid models, showing how STAT3 and MEK/ERK pathways are also solicited in crosstalk between tumour cells and GCAFs for the maintenance of tumour integrity [76]. Loss of Trefoil Factor 1 (TFF1) is observed in human GCs, linked to STAT3 nuclear localisation and to pro-inflammatory phenotype and gastric lesions in mice. TFF1-knockout mice demonstrate a nuclear localisation of p-STAT^Y705^ along with an overexpression of STAT3 target genes and on the contrary, the reconstitution of the protein in GC cell lines and organoids from TFF1-knockout mice decreases p-STAT^Y705^ showing the role of JAK/STAT pathway in GC. TFF1 blocks the dimerization of IL6Rα and GP130 subunits and thus obstructing the binding of IL6 on its receptor and its pro-inflammatory and pro-tumorigenic consequences [77]. Furthermore, Galectin-1 (Gal-1), a β-galactosidase-binding protein, is highly expressed in GCAFs and is involved in GC progression and metastasis through the activation of signalling cascades like the Hedgehog (Shh) pathway involved in the EMT process. Gal-1 from GCAFs, binds to a carbohydrate structure in β-1 integrin and induces EMT, cell migration and invasion by regulating Glioma-associated oncogene-1 (Gli-1). High expressions of Gal-1 and Gli-1 are correlated to poor prognosis of patients [78]. GCAFs also induce the expression of Discoidin domain receptor 1 (DDR1) in GC cells leading to increased tumorigenesis both in vitro and in vivo [79]. Communications between the stroma and cancer cells are not unidirectional. Indeed, cancer cells also produce factors which modulates the stroma. GC cells produce exosomes which induce pericytes proliferation, migration, and expression of CAFs markers showing that cancer cell-derived exosomes are involved in the transition of pericytes into CAFs [80]. GC cell-derived exosomes are able to induce BMP (Bone Morphogenic Protein) transfer and activate PI3K/AKT and MEK/ERK in pericytes transforming them into GCAFs. The use of BMP signalling pathway inhibitor Noggin reverses the tumour-exosome-induced pericyte-CAF transition and decreases the expression of CAFs markers by pericytes [80]. *Helicobacter pylori* c*an* also participate to stroma-induced gastric carcinogenesis. In presence of the bacterium, gastric fibroblasts are activated into GCAFs with an increased secretion of TGF-β and conditioned medium from *H. pylori*-activated GCAFs prompt*s* EMT-like phenotype in rat gastric epithelial cells RGM-1 [81].

Tumour associated macrophages (TAMs) can be of 2 phenotypes, M1 and M2, M2 being the cancer cell-activated form of TAMs, essential for their role in tumour growth and progression [70]. GC tissues are found to be composed mostly of M2 TAMs, which when isolated promote migration and invasion of GC cells [73]. Interestingly, recent single cell tumour microenvironment genomic characterisation demonstrate that GC TAMs genetic signatures do not match those of known M1 or M2 macrophages [82]. M1 and M2 known distinguishing genetic markers are co-expressed in the same macrophage cluster. Nevertheless, genes distinguishing these different TAM clusters seem to be related to the HSP family, *THBS1*, MMPs family, *CCL20*, *CCL18* and *CCL3* chemokines among others, as well as cell cycle-dependent genes. In addition, these TAMs seem to differ from PBMC monocytes but conserve some similarities with normal tissue macrophages [82]. Moreover, TAMs abundance in GC microenvironment is associated with a decrease in number and a dysfunction of Natural Killer (NK) cells, which can be reversed by TGFβ1 blockade [83]. In so doing, TAMs promote tumour immune response in GC. Furthermore, TAM programmed cell death protein 1 (PD1) expression increases as GC progresses in mice and this can also be found in primary human cancers [84]. PD1 immune checkpoint receptor is normally present on activated T cells to induce their immune tolerance. TAM PD1 expression is correlated to less phagocytosis of tumour cells which express PD1 ligand, PD-L1, as immune escape mechanism [84]. 

In relation with immunosuppression mechanisms, Regulatory T cells (Tregs) seem to be abundant in gastric TME [82,85]. Sathe et al. single cell analysis of gastric TME puts into light two different Treg classes marked by different expression level of proliferation-associated genes.

Dendritic cells (DCs) can also reflect GC patient prognosis. Liu et al. showed that different subsets of DCs, plasmacytoid DCs (pDCs) and myeloid CD1+ DCs (mDC1s) are increased in GC patients peripheral blood compared to healthy controls [86]. Also, high level of pDCs seems to correlate with GC stage progression while mDC1 level does not change. Furthermore, different DC subtypes are found according to GC stages. Early GC T stages present high peripheral DC2 number and tumour infiltrating DC1/DC2 ratio, while low level of tumour infiltrating DC2s and high DC1/DC2 ratio are observed at N0 GC stage [87].

Stromal R-spondin, produced by gastric myofibroblasts normally controls gastric epithelial cell and gland homeostasis by activating the proliferation of Lgr5+ gastric stem cells. *H. pylori* increases the expression of R-spondin 3 and causes the hyperproliferation and hyperplasia of the gastric gland [88]. Interestingly, R-spondin 3, required physiologically for the *m*aintenance of Lgr5+ cell identity, induces differentiation into a secretory phenotype and, according to a recent study, allows these cells to develop defence mechanisms against bacterial infection. In presence of *H. pylori*, R-spondin 3 enables Lgr5+ cells to secrete anti-microbial protein Interlectin-1 (Itln1) which binds to the bacteria and agglutinates them. R-spondin-Lgr5 signalling in the stomach thus protects the gland base against infectious agents [89].

Finally, an acetylcholine-Nerve Growth Factor (Ach-NGF) axis can activate GC niche and promote the disease. Nerves within the gastrointestinal niche regulate both normal and neoplastic stem cell dynamics. Dclk1+ tuft cells and nerves, main Ach source in the gastric mucosa, induce a cholinergic stimulation of NGF expression, which, when overexpressed, expand enteric nerves, and promote carcinogenesis though YAP function [90].

### 3.4. A Microbiome Disease

Although the stomach was once thought to be a sterile environment, it is now known to house many bacterial species, leading to a complex interplay between *H. pylori* and other residents of the gastric microbiota. Data now show that the stomach harbours a large and diverse bacterial community with colonisation densities ranging from 10^1^ to 10^3^ colony forming units/g [91]. Moreover, recent advances in molecular techniques and computational analysis have provided evidence that the complex microbiota colonising the gastric epithelium in combination with *H. pylori* might influence gastric homeostasis and disease [92]. Currently, there are few studies that have examined differences in microbial composition and outcomes of GC. One of the key steps in the histological progression to intestinal type GC is the development of atrophic gastritis, a condition that predisposes the stomach to an increase in gastric pH due to loss of parietal cells and overgrowth of non-*Helicobacter* microbiota [14]. This stomach hypochlorhydria facilitates colonization by bacteria and may promote the progression towards gastric cancer, confirming that microbiota can influence gastric carcinogenesis [14].

In 2011, a study showed that in the INS-GAS mice model of spontaneous gastric cancer, gastric lesions take 13 months longer to develop when the mice are germ-free than when they are Specific Pathogens Free (SPF) [93]. Furthermore, the authors showed that *H. pylori*-mono association accelerates gastritis and GIN but causes less-severe gastric lesions and delays the onset of GIN compared to *H. pylori* infection of INS-GAS mice with complex gastric microbiota. This data proves that microbiota can influence gastric carcinogenesis and has been confirmed with further works showing that gastric colonisation with a restricted commensal microbiota can replicate the promotion of neoplastic lesions by diverse intestinal microbiota [94].

In addition, in a recent study describing GC patient microbiota and comparing it with that of chronic gastritis patients, it has been shown that GC microbiota is characterised by reduced microbial diversity, decreased abundance of *Helicobacter* and by the enrichment of other bacterial genera, mostly intestinal commensals. Furthermore, an analysis of the functional features of the microbiota revealed that GC patients’ microbiota is compatible with the presence of a nitrosating microbial community. Thus, patients suffering from gastric carcinoma have a dysbiotic microbial community with genotoxic potential, which differs from the gastritis patients [95].

Two recent studies have described and analysed the gastric microbiome of different types of patient. A Taiwanese study profiled gastric epithelium-associated bacterial species in patients at different stages of the Correa cascade (gastritis, intestinal metaplasia and gastric cancer) [96] and found that the overall microbiota composition is similar between patients with gastritis and those with intestinal metaplasia. In GC patients, *H. pylori* is indeed more frequent and more abundant, corresponding to a cancer-specific signature, but, *Clostridium*, *Fusobacterium* and *Lactobacillus* species are also frequently abundant in those patients, though their role in gastric carcinogenesis is still not proven. Following this work, another study compared the gastric microbiota of patients presenting GC to controls having a functional dyspepsia [97]. The authors showed that several bacterial taxa are enriched in GC patients, particularly pro-inflammatory oral bacterial species, lactic acid producing bacteria and bacteria producing short fatty acids. These results are for the moment only descriptive and *H. pylori* still remains the only bacterium directly implicated in gastric carcinogenesis.

Although several works have shown that gastric microbiota can influence gastric carcinogenesis, there is for now no report of a potential microbiota-based therapeutic or prognosis strategy in GC context. Further work in this direction is thus warranted. 

## 4. Biomarkers and Therapeutic Strategies

### 4.1. Biomarkers and Targeting

Gastric cancer treatment still consists of surgery and additional adjuvant or neoadjuvant radio- and chemotherapies. Early tumours can be excised endoscopically, and surgical resection remains the only curative treatment with unfortunately, a high number of relapse cases. Conventional chemotherapies consisting of leucovorin, oxaliplatin, docetaxel and 5-FU are given as perioperative chemotherapy or as palliative chemotherapy for patients having advanced metastatic disease [1,98]. Nevertheless, GC patient 5-year survival rate remains less than 5% for advanced unresectable or metastatic cases (80% of patients at diagnosis, except for Far East countries) [1].

The need for personalized targeted treatment is urgent and currently, only few targeted therapies exist. Trastuzumab (Herceptin, Roche, Germany) targets HER2+ GC and is used as first-line treatment for patients with tumours overexpressing HER2, accounting for less than 30% GC cases [98]. Ramucirumab (Cyramza, Lilly, USA) is another targeted treatment for advanced refractory GC cases. This antibody, used alone or in combination with paclitaxel, targets vascular endothelial growth factor receptor 2 (VEGFR2) thus inhibiting angiogenesis and increasing survival. However, these targeted therapies do not cater for all the different GC types having diverse aggressiveness levels and response to treatment [4,8,10].

The multiple molecular classification systems that exist indeed influence endoscopic or surgical choices but are not enough to guide precision individual treatment decisions which are urgently needed. External factors like *H. pylori* infection are being catered for through the multiple eradication strategies which indeed seem to be efficient [18]. Unfortunately, in patients where the disease has evolved, the establishment of genetic as well as cell phenotypic changes like the acquisition of resistant CSC and invasive properties make the tumours resistant to conventional therapies [98]. In addition, patients are diagnosed at late stages of the disease, and proper diagnosis markers development could help detect GC at earlier stages, allowing proper patient care.

Carbohydrate antigen 19-9 is a commonly used biomarker in GC allowing recurrence prediction when in combination with other tumour markers [99]. Molecules directed against Tyrosine Kinase receptors like cetuximab, rilotumumab or dovitinib could be suitable against GC cases overexpressing their respective targets. Clinical trials for antibodies against FGFR2 (for FGFR2 overexpressing GC) and EGFR (Nimotuzumab; for EGFRhigh GC) are ongoing. Similarly, mTOR pathway inhibitor Everolimus is being tested for patients with MSI type GC often having activating mutations in members of this pathway [98,100].

Studies are more and more trying to understand mechanisms underlying gastric carcinogenesis, may it be induced or not by its principal risk factor *H. pylori*. In so doing, these studies explore molecules which are either over- or under-expressed in GC compared to non-tumorous tissues and these particular signatures could indeed be used in GC diagnosis and prognosis (Table 1). 

Hepatoma-derived growth factor (HDGF), overexpressed in GC patients and after *H. pylori* infection, seems to participate in *H. pylori*-induced neutrophils recruitment, gastritis and gastric carcinogenesis. HDGF level is also high in patients with intestinal metaplasia and is associated with low survival. Targeting HDGF decreases neutrophil infiltration and inflammatory responses induced by *H. pylori* infection, making HDGF a potential therapeutic target for *H. pylori-*induced GC treatment [101]. 

Several clinical trials involving antibodies targeting the HGF receptor c-Met were also started but did not give encouraging results. However, positive results were reported from a Phase I trial with a MET specific small molecule AMS 337 showing 1 complete response and 4 partial responses out of 10 patients. MET amplification is observed in 4% GC patients only and bad trial results might be related to an incorrect selection of patients prior to trial since usual immunohistochemistry selection is not the most sensitive one [100]. Unfortunately, resistance to small molecule inhibitors of MET kinase is quite common. In this context, Andres et al. used the designed ankyrin repeat protein (DARPin) technology and generated MET-binding DARPins covering extracellular epitopes of MET, thus creating sets of bi-paratopic fusion proteins which efficiently inhibited MET kinase activity and downstream signalling [102]. This strategy using proteins having two paratopes caused receptor downregulation and inhibition of MET induced GC cells proliferation and seems to be an interesting strategy for targeting receptors in therapy [102]. Furthermore, Zhang et al. showed that exosomes could be used to deliver c-Met siRNA to GC cells thus inhibiting migration, invasion and inducing apoptosis in vitro and enhancing sensitivity to cisplatin in vitro as well as in vivo. The exosomes used were isolated from human embryonic kidney epithelial cell line, HEK293T, transfected with si-c-Met [103]. 

Bone marrow stromal cell antigen 2 (BST2) or CD317, playing a crucial role in antiretroviral defence in innate immune response, is also overexpressed in GC tissues compared to contiguous non-tumorous tissue and is correlated to tumour stage and lymphatic metastasis [104]. BST2 siRNA inhibited GC cell proliferation and motility and induced apoptosis. Furthermore, NF-κB pathway seemed to be involved in the pro-tumour function of BST2. BST2 targeting could be another possible strategy for GC treatment [104]. 

Moreover, the cell surface protein CEACAM6 has been identified as a potential endoscopic marker of early GC. CagA+ *H. pylori* induces this protein expression, which is highly present in early GC and pre-malignant lesions. Fluorescently tagged antibody against CEACAM6 was proposed to mark GC tissue for visualisation, using commercially available endoscopic methods, for use in diagnosis [105]. 

### 4.2. GCSC Targeting

Though the major tumour mass is found to be chemo-sensitive and would respond to therapies targeting GC subtypes or immerging immunotherapies, part of the cells comprising the tumour mass, GCSCs, are believed to resist treatments and to be at the origin of cancer relapse and metastasis [98,106]. GC molecular signatures are more and more being explored to propose diagnosis and prognosis markers and to try to target proteins or signalling pathways having a role in the maintenance of these GCSCs. 

Several drugs have been evaluated for repositioning as anti-CSC strategy (Table 2). Verapamil, a calcium antagonist allowing the inhibition of calcium-dependent channels and used to treat angina pectoris and cardiac arrythmias, was found to block efflux mechanisms of GCSCs which normally allow these cells to evacuate chemotherapy drugs through calcium-dependent channels. The use of verapamil thus sensitized GCSCs to conventional chemotherapies [55]. Tretinoin, also known as all-trans retinoic acid (ATRA), used for the treatment of acne and acute promyelocytic leukaemia due to its pro-differentiation properties, was able to force differentiation of GCSCs thus impacting their tumorigenic self-renewal capacity [107]. Furthermore, metformin, used for the treatment of type 2 diabetes to decrease insulin resistance and hepatic neo-glucogenesis, showed efficient anti-GCSCs effects by targeting the metabolism of these cells [108]. Drug repositioning was also used as strategy to target signalling pathways exacerbated in CD44+ GCSCs. Buparlisib (BKM120, a pan-class I PI3K inhibitor), first line treatment of metastatic head and neck epidermoid carcinomas, was found to decrease GCSC tumorigenic properties in vitro and to decrease GC metastasis in vivo [109]. Verteporfin, an FDA-approved drug for age-related macular degeneration, was repositioned in the GCSC context for its capacity to decrease YAP/TEAD transcriptional activity, cell proliferation, CD44 expression and number of tumorsphere-forming CD44+ALDHhigh GCSCs in vitro. Verteporfin also inhibits tumour growth in vivo [59]. In addition, residual tumour cells were unable to form new tumorspheres in vitro confirming the decrease of the in vivo CSC pool after verteporfin treatment [59]. Recently, LIF cytokine treatment was found target GCSCs through LATS2-induced repression of YAP1/TAZ/TEAD oncogenic activity [60]. 

Most GCSC-targeting based strategies are directed against CD44+ cells (panCD44) which is ubiquitously expressed in non-tumorous cells even though its expression is exacerbated in GCSCs. Thus, using anti-panCD44 as GCSC target could cause non-specificity problems. CD44 variants, which seem to be less expressed in non-tumorous tissue compared to panCD44, are an interesting alternative for GCSC targeting. Studies have shown the increased expression CD44v8-10 and CD44v9 in GCSCs but also their functional role in the maintenance of this chemo-resistant cell population [63].

Vismodegib, an antagonist of the Shh signalling pathway has been associated with leucovorin, 5-FU and oxaliplatin in the treatment of GC. It binds to SMO, in GCSCs, thus preventing the downstream activation of GLI family of transcription factors and inhibiting Shh signalling. CD44+ GC cells present an overexpression of Shh pathway proteins linked to low patient survival, which was improved after the combined treatment [106]. Similarly, Napabucasin, a STAT3 inhibitor, repressing CSC self-renewal and inducing cell-death in GCSCs by targeting STAT3 was tested for GCSC targeting. Napabucasin was used in combination with paclitaxel in patients with advanced tumours and showed an anti-tumour activity. These studies show that combining chemotherapy to targeted strategies seems to be an interesting approach for anti-GCSC directed GC treatment [106].

High production and low elimination of reactive oxygen species (ROS) by the organism is another cancer risk factor. ROS are tightly controlled under physiological conditions through antioxidant mechanisms since excessive ROS in cells can lead to DNA damage. This property of ROS makes them interesting in therapeutic strategies in order to induce cancer cell damage and death. Unfortunately, cancer cells specially CSCs seem to possess defence capacities against ROS making them resistant to therapies. CD44v9 found in GCSCs can activate xCT, glutamate-cystine exchange transporters, which help increase levels of intracellular reduced glutathione (GSH) and contribute to CSC survival in ROS-high environment [63]. Exogenous CD44v9 expression in cells increase resistance to 5-FU, a chemotherapy agent using ROS production mechanism to kill cancer cells. Inhibition of xCT involved in the anti-ROS mechanism of GCSCs enhances 5-FU anti-GC efficiency [110].

Apart from their use in diagnosis, miRNA targeting, or overexpressing strategies can also be considered as GC therapeutic solutions. miR-7-5p normally exerts its tumour-suppressing effect, through the downregulation of its target genes SMO and HES1 (members of the Shh and Notch signalling pathways respectively), which is lost in GCSCs where this miRNA is under-expressed. So, targeting SMO and HES1 in GC could serve as target for GCSCs [106]. Seed-targeting locked nucleic acid (LNA) can be used as specific miRNA inhibitors and target miR-372/373 thus decreasing GC cell growth and targeting GCSCs [111].

### 4.3. Liquid Biopsies as Biomarkers

Certain subpopulations of GCSC, metastasis-initiating cells (MIC), overexpress MMPs, which are physiologically involved in extracellular matrix breakdown and promote invasion and metastasis. MMP10, MMP15 and MMP9 are found to be increased in GC and to correlate with poor patient prognosis [38,112] During the metastasis process, tumour cells originating from primary tumour or metastases can be found circulating in blood, either single or in clusters. New techniques have allowed the detection of these circulating tumour cells (CTCs) which are characteristic of disease progression and metastatic processes and can be used as surveillance markers [112]. CTCs analysis allows the detection of early metastasis stages and are used to identify patients fit for chemotherapy after primary tumour resection. CTC properties can be evaluated to see whether they carry CSC or EMT-like properties allowing better prognosis prediction. In GC, the presence of CD44+ GC CTCs has been correlated to tumour metastasis and relapse. Some studies show that circulating cell-free DNA are more sensitive than CTCs for diagnosis and prognosis. Circulating tumour DNA (ctDNA) originates from primary tumours or metastases and allows more specific diagnosis of patients as well as the assessment of therapeutic response [112]. 

In addition, miRNA differential expression in GC and high efficiency of circulating-miRNA detection assays make them interesting non-invasive biomarkers for GC [112]. miR-192-5p and miR-9-5p are highly decreased in GC tissues compared to non-tumorous gastric tissues and can thus be candidate for GC diagnosis [113]. Moreover, miR-9-5p and a combined miRNA group (miR-9-5p + miR-9-3p + miR-433-3p) were found to distinguish chemo-resistant GC patients from chemo-sensitive ones [114], confirming the interest of miRNA as novel non-invasive diagnostic tool in GC. 

Furthermore, exosomes which are small vesicles produced by cells carry RNAs and miRNAs which remain protected from degradation when exposed to RNAses. Cancer cells or CAFs use exosomes to communicate and exchange material. These vesicles are a great promise for GC diagnosis and prognosis. Studies show that miRNA can be identified in serum-circulating exosomes, allowing the staging of patients. For example, exosomes containing miR-29s are found to play a suppressive role in the growth of peritoneal-disseminated tumour cells and are under-expressed in patients with peritoneal metastases [115]. Low expression of this miRNA in exosomes is correlated to bad prognosis of patients. The use of exosomes as biomarkers and even in therapy (as described above) seems to be an interesting strategy which will surely evolve in the next few years [116]. In addition, long non-coding RNAs (lncRNAs) can also be detected in exosomes. lncRNA MIAT, for example, has been described as overexpressed in GC patients and associated to lower survival rates. Serum exosomal MIAT can be detected, decreases post-treatment and is highly increased in patients suffering from GC relapse. This serum exosomal level of MIAT could thus be used to monitor GC progression using liquid biopsies [117].

### 4.4. Immunotherapy

Immunotherapy is becoming a promising anti-cancer strategy for many cancers. PD-L1 can be used as biomarker of GC involving immune checkpoint escape. Nivolumab (Opdivo, Bristol-Myers Squibb, USA), an antibody targeting PD1 and disrupting its interaction with PD-L1 and PD-L2 and increasing T lymphocytes anti-tumoral activity has been recently approved in Japan as third-line treatment for unresectable or recurrent GC patients having already undergone 2 chemotherapeutical strategies. Furthermore, US Food and Drug Administration (FDA)-approved anti-PD-L1 antibody Pembrolizumab (Keytruda, Merck & Co., USA) is used for treating PD-L1+ recurrent or advanced GC patients with 2 or more prior lines of therapy [98,118]. Although PD1 and PD-L1 inhibitors seem to improve the outcome of a small group of GC patients, the way to identify the patients that would respond still needs to be improved. Trials indeed show heterogenous responses and even the way PD-L1 is used as biomarker needs to be considered since qualitative or quantitative PD-L1 expression-based selected patients will not have similar responses [98,118].

Finally, the use of CAR T-cell therapy with T-cells binding CSC-specific antigens could be an interesting path to follow. These cells could specifically target GCSCs and eliminate them. Unfortunately, GCSC markers are not specific since they are ubiquitously expressed in other non-pathologic cells and imply a non-specific targeting of tumours even if these markers are more intensively expressed in tumour cells. Despite all this, two CAR T-cell therapies have been approved for the treatment of children with acute lymphoblastic leukaemia and adults with advanced lymphomas, showing the positive evolution of this field. In GC, several in vitro and in vivo trials can be found in literature. HER2 CART-T cells showed positive response in vitro and persisted in blood circulation, specifically travelled to and accumulated in HER2 overexpressing tumours and contributed to their regression in human GC xenograft models [119]. Another study describes the production of anti-GC cells monoclonal antibody mAb-3H11 by the hybridoma technique with spleen cells from mice immunised with five different human GC cell lines. mAb-3H11 was selected for its high specificity for GC cells and no reaction with normal cells. A single-chain variable fragment (scFV) of mAb-311, displaying the same reaction as the whole antibody, was used to design CAR-T cells which were able to kill tumour cells in vitro and GC cell lines as well as patient-derived xenograft tumours in vivo [119]. The same in vitro and in vivo anti-tumoral effects were observed when Folate receptor 1 (FOL1), overexpressed in more than one third GC patients, coupled to CAR-T cells were used [120]. Among the 38% clinical trials being performed for CAR-T cells on solid tumours, only 2.96% account for GC with 12 registered clinical trials (ClinicalTrial.gov) distributed between China and the USA. These trials target different antigens (EPCAM, MUC1, CEA, HER2, Mesothelin, BPX-601 and EGFR) and most are still in the recruiting stage [121]. High toxicity mainly due to cytokine release syndrome, one of the main side effects which occurs due to the rapid and high activation of numerous cytokines, is what restrains this field for now [121]. More research deserves to be carried out with the perspective of finding even more specific targets for GCSCs and limiting the toxicity [98].

## 5. New Experimental Models

Research issues are in constant evolution and so must be the strategies used to understand and resolve them. *Helicobacter pylori* infection has been for long known to be a major GC risk factor and the studies relating GC and this bacterium mostly depends on the use of transformed cell lines as infection models. This artificial approach can be criticised for its lack of pragmatism in terms of signalling pathways study, crucial in this context. Barker et al. found, using lineage tracing, that Lgr5+ cells were the multipotent stem cells responsible for the long-term self-renewal of the gastric epithelium and were among the first to generate organoids resembling mature pyloric epithelium using single Lgr5+ cells in vitro [122]. Afterwards, a new gastric primary cell culture system was developed for modelling *H. pylori* infection in vitro. Using gastric glands isolated from healthy human stomach tissue and growth factors-supplemented Matrigel, the authors were able to grow 3-dimensional (3D) spheroids which can differentiate into gastric organoids after the withdrawal of Wnt3A and spondin-1 from the culture medium leading to the formation of cultures of polarised epithelial cells when transferred into 2D [123]. However, these structures offer only suboptimal conditions for studying consequences of bacterial infection due to their closed spherical shape. A recent study proposes a “mucosoid culture” model using antrum-derived gastric glands and air-liquid interface culture technique. The polarised epithelial monolayers formed secrete mucus at the apical surface, reproduce normal human gastric epithelium and can even differentiate into a base-like gland phenotype under the influence of Wnt signalling [124].

Studying gastric carcinogenesis also involve the use of proper in vivo models reflecting the underlying process. The major limitation of mouse models using *H. pylori* infection or carcinogens to induce gastric carcinogenesis is that they only rarely develop in situ carcinomas since most of the lesions are pre-neoplastic ones. In addition, these models do not metastasize or invade like in humans. Likewise, subcutaneous xenograft of PDX cells reflect the heterogeneity of patient tumours but still do not metastasize to distant organs [55]. In a recent study, *Giraud* et al. developed orthotopic PDX models in which patient-derived GC cells were xenografted directly into the stomach wall of immunodeficient mice and led after 8 weeks to distant metastases [109]. In these pre-clinical models, luciferase-encoding GC cells were traced all through the in vivo experiment allowing the monitoring of primary tumour establishment and kinetic of GC cells spread and metastasis development. Using these models, the authors showed that Buparlisib treatment significantly inhibited GCSC properties in vitro and reduced the number of distant metastases in vivo when the treatment was done in the metastases starting time-lapse determined by the model [109]. This preclinical mouse model of metastatic GC represents a major advance to study anti-metastatic efficiency of new GCSC-based therapies.

## 6. Conclusions

Gastric cancer, as many other cancers, is a complex and multifactorial disease. *H. pylori* remains the main cause for GC, despite the participation of other extrinsic and intrinsic factors. Gastric tumours are highly heterogenous both at intra-tumoral and inter-tumoral levels, with different histological and molecular subtypes and cellular hierarchy within the tumour as well as in the TME composition. The complexity of this disease makes it such that despite research advances and all the highlighted potential biomarkers and therapeutical strategies, there are still only few targeted strategies like Trastuzumab, Ramucirumab and anti-PD1/PD-L1 immunotherapies used in clinic [98]. A better understanding of this gastric disease’s cellular, molecular, and infectious processes, at the basis of this tumour heterogeneity, is critical for the development of other diagnosis and therapeutic strategies against GC. 

## Figures and Tables

**Figure 1 ijms-22-03418-f001:**
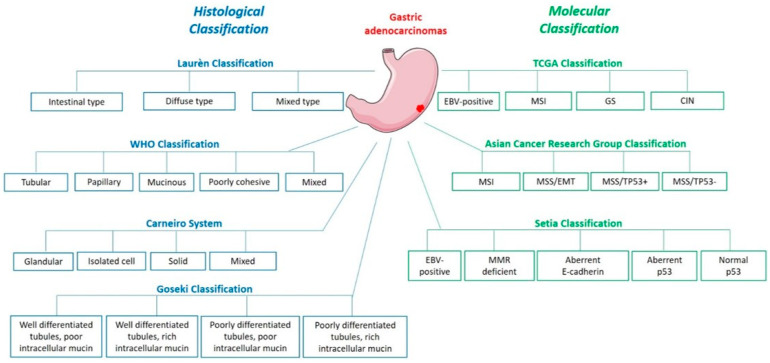
Gastric adenocarcinoma histological and molecular classifications.

**Figure 2 ijms-22-03418-f002:**
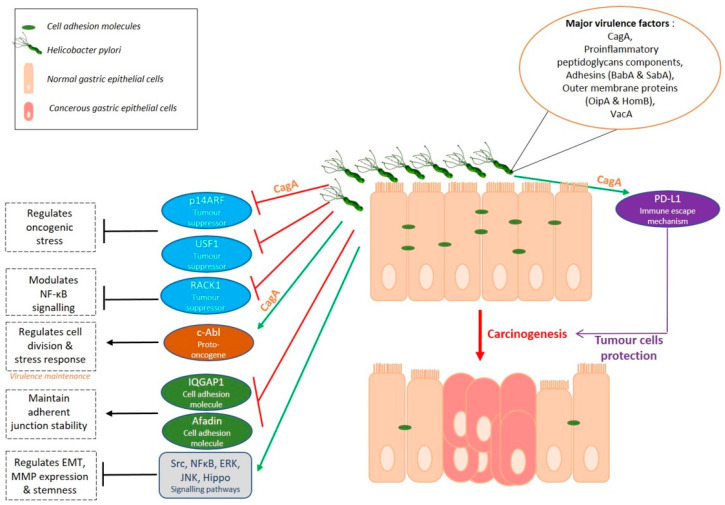
Gastric carcinogenesis: a *Helicobacter* disease.

**Figure 3 ijms-22-03418-f003:**
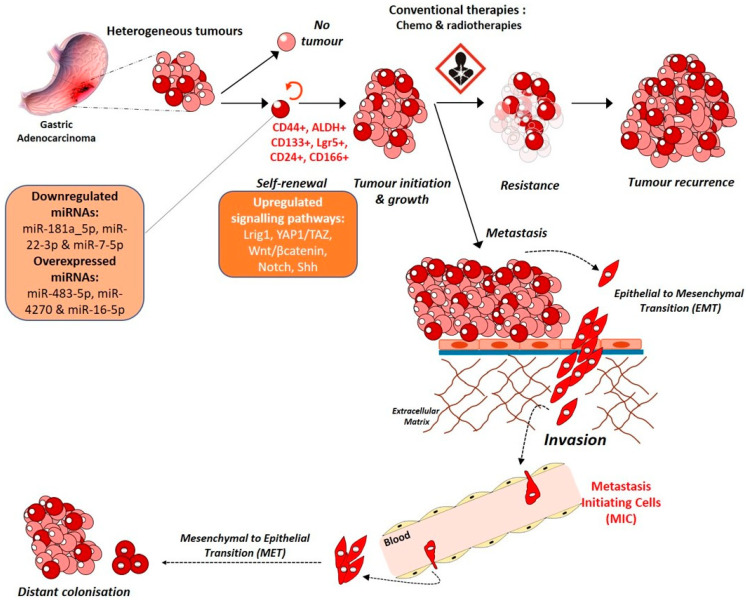
Gastric carcinogenesis: a stem cell disease.

**Figure 4 ijms-22-03418-f004:**
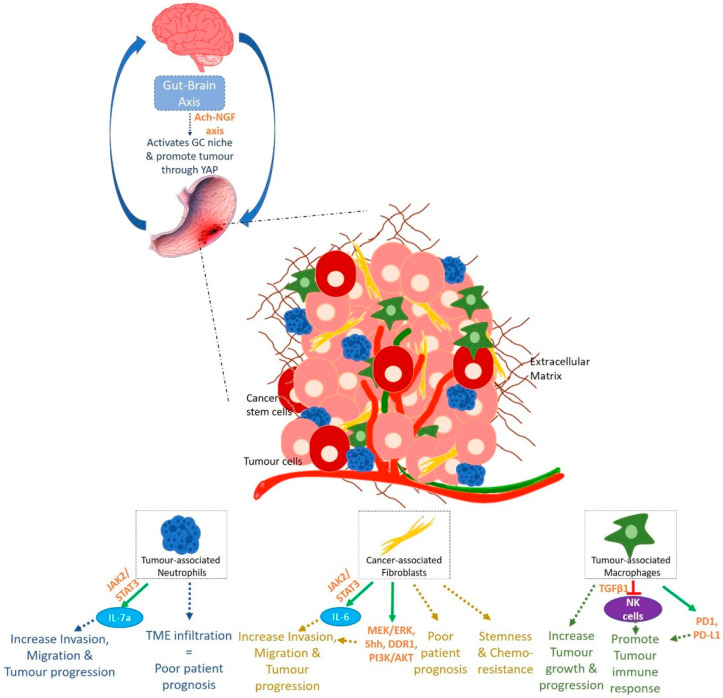
Gastric carcinogenesis: a microenvironment disease.

**Table 1 ijms-22-03418-t001:** Summary of molecules that can be used as potential biomarkers and therapeutic targets for GC.

Molecule/Strategy	Targets	Known/Tested Use in GC	References
Anti-CA19-9 antibodies	CA19-9	CA19-9-positive GC biomarker for diagnosis	Matsuoka et al. 2018 [99]
Cetuximab	EGFR	Potential targeted therapy against tyrosine kinase receptors	Carrasco-Garcia et al. 2018 [98], Apicella et al. 2017 [100]
Rilotumumab	HGF	Potential targeted therapy against tyrosine kinase receptors	Carrasco-Garcia et al. 2018 [98], Apicella et al. 2017 [100]
Dovitinib	VEGFR-1/2/3; PDGFR-β; FGFR1/2/3	Potential targeted therapy against tyrosine kinase receptors	Carrasco-Garcia et al. 2018 [98], Apicella et al. 2017 [100]
Anti-FGFR2 antibodies	FGFR2	Under clinical trial for FGFR2 overexpressing GC	Carrasco-Garcia et al. 2018 [98], Apicella et al. 2017 [100]
Nimutuzumab	EGFR	Under clinical trial for EGFRhigh GC	Carrasco-Garcia et al. 2018 [98], Apicella et al. 2017 [100]
Everolimus	mTOR pathway	Under clinical trial for MSI type GC with activating mutations of mTOR pathway members	Carrasco-Garcia et al. 2018 [98], Apicella et al. 2017 [100]
Anti-HDGF antibodies	HDGF	Potential prognostic marker & target of *H. pylori*-induced GC	Chu et al. 2019 [101]
AMS 337	c-Met	Positive results in Phase I clinical trial	Andres et al. 2019 [102]
MET-binding DARPins	c-Met kinase activity	Potential receptor targeting strategy	Andres et al. 2019 [102]
Exosomes-delivered c-Met siRNA	c-Met	Potential use as therapy in combination with chemotherapy	Zhang et al. 2020 [103]
BST2 siRNA	BST2	Inhibits GC cell proliferation and motility – potential anti-GC therapy	Liu et al. 2018 [104]
Anti-CEACAM6 antibodies	CEACAM6	Potential endoscopic marker for early GC diagnosis	Roy et al. 2016 [105]

**Table 2 ijms-22-03418-t002:** Summary of potential Drug and/or Molecules for the targeting of GCSCs.

Drug/Molecules	Target/Effects	Known Effects in GC	References
Verapamil *(in combination with chemotherapies)*	Inhibit calcium-dependent channels	Blocks drug efflux mechanisms of CD44+ALDH+ GCSCs and prevents resistance to conventional therapies	Nguyen et al. 2017 [55]
Tretinoin	FDA-approved drug for topical treatment of acne vulgaris; pro-differentiation properties	Forces differentiation and decreases tumorigenic properties of CD44+ALDH+ GCSCs	Nguyen et al. 2016 [107]
Metformin	FDA-approved drug for first-line treatment of type 2 diabetes; decreases insulin resistance and hepatic neo-glucogenesis	Decreases tumorigenic properties of CD44+ GCSCs by targeting EMT and metabolism modulation	Courtois et al. 2017 [108]
Buparlisib	Pan-class I PI3K inhibitor	Decreases CD44+ GCSC tumorigenic and metastatic capacity	Giraud et al. 2019 [109]
Verteporfin	FDA-approved drug for age-related macular degeneration – inhibits Hippo effector YAP/TAZ-TEAD interaction	Decreases CD44+ALDH+ GCSC tumorigenic properties through Hippo pathway oncogenic effectors inhibition	Giraud et al. 2019 [59]
LIF cytokine	Pro-differentiation properties	Decreases CD44+ALDH+ GCSCs tumorigenic properties by inducing Hippo tumour suppressor kinases activity	Seeneevassen et al. 2020 [60]
Vismodegib *(in combination with chemotherapies)*	FDA-approved drug for recurrent locally advanced and/or metastatic Basal Cell carcinoma; antagonist of the Shh signalling pathway	Improves patient survival in combination with chemotherapies by targeting CD44+ GCSCs having high Shh pathway activity	Bekaii-Saab et al. 2017 [106]
Napabucasin *(in combination with chemotherapies)*	FDA-approved as orphan drug for treatment of gastroesophageal junction cancer; STAT3 inhibitor	Decreases GCSCs tumorigenic properties in combination with paclitaxel in patients with advanced tumours	Bekaii-Saab et al. 2017 [106]
Glutamate-cystine exchange transporters inhibitor (xCT)	xCT inhibition	Sensitizes GCSCs to 5-FU conventional therapy by blocking xCT anti-ROS mechanisms	Miyoshi et al. 2018 [110]

## Data Availability

Data is contained within the article.

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
