# Peer review of "Gastric Cancer: Advances in Carcinogenesis Research and New Therapeutic Strategies"

_ijms, 2021, doi:10.3390/ijms22073418_

Round 1

Reviewer 1 Report

In the manuscript »Gastric cancer: advances in carcinogenesis research and new therapeutic strategies« the authors provide a comprehensive review on pathogenesis of gastric cancers as well as emerging treatment options. In the first part of the manuscript they explain clinically used histological classifications and relevant epidemiological data as well as risk factors for the disease. In the carcinogenesis section they describe very detailed the important role of H. pylori in the development of gastric cancer as well as potential role of cancer stem cells, tumor microenvironment as well as emerging insights in the role of other residents of gastric microbiota. Based on new insights on pathophysiology of the disease they elaborate the current systemic treatment options as well as potential new therapies. They also nicely explain the possibility of new biomarkers based on the liquid biopsy concept. The role of immunotherapy is also explored including the new concept of CAR T-cell therapy. Some new experimental models are also briefly mentioned at the end of the manuscript. The conclusions are sound and clear, the references are relevant and updated.

Author Response

Our point-by point response to the reviewer's comments are in bold under each comment.

Reviewer 1

In the manuscript »Gastric cancer: advances in carcinogenesis research and new therapeutic strategies« the authors provide a comprehensive review on pathogenesis of gastric cancers as well as emerging treatment options. In the first part of the manuscript they explain clinically used histological classifications and relevant epidemiological data as well as risk factors for the disease. In the carcinogenesis section they describe very detailed the important role of H. pylori in the development of gastric cancer as well as potential role of cancer stem cells, tumor microenvironment as well as emerging insights in the role of other residents of gastric microbiota. Based on new insights on pathophysiology of the disease they elaborate the current systemic treatment options as well as potential new therapies. They also nicely explain the possibility of new biomarkers based on the liquid biopsy concept. The role of immunotherapy is also explored including the new concept of CAR T-cell therapy. Some new experimental models are also briefly mentioned at the end of the manuscript. The conclusions are sound and clear, the references are relevant and updated.

Thank you very much for your review and kind comments.

Reviewer 2 Report

The manuscript covers many aspects of GC, possible biomarkers and different therapeutic approaches. The authors have written well but it lacks systematic organization. Use of table and graphical presentation will be helpful for readers.

Few points to consider:

  1. H. Pylori story is overwritten. Line 333-355, 357-362: Remove it from stem cell discussion. It may fit in the Helicobacter disease paragraph.
  2. In section 3.4 what are the other microorganisms involved in GC except H. Pylori? The authors have mentioned Clostridium, Fusobacterium and Lactobacillus but they aren’t elaborated properly. Rewrite the 3.4 section.
  3. Section 4.1: A table must show all the biomarkers and specific approaches to detect GC. Line 580-600: this stanza may be better suitable for microenvironment paragraphs. I don’t find HDGF as a biomarker in GC from the manuscript.
  4. First write about biomarkers then come to targeting approaches or therapy. Liquid biopsy biomarkers should come first then GCSC targeting.
  5. Conclusion should be rewritten with specific biomarker and therapeutic approaches which are mostly used for GC at present time. Which approaches are better and should be considered in future for GC treatment?

Author Response

Our point-by point response to the reviewer's comments are in bold under each comment.

Reviewer 2

The manuscript covers many aspects of GC, possible biomarkers and different therapeutic approaches. The authors have written well but it lacks systematic organization. Use of table and graphical presentation will be helpful for readers.

Thank you for your comments. In fact, we had done figures recapitulating most of the sections, which were not included in the text but had been submitted initially. We have added the different figures to the text.

Few points to consider:

  1. H. Pylori story is overwritten. Line 333-355, 357-362: Remove it from stem cell discussion. It may fit in the Helicobacter disease paragraph.

Thank you very much for your suggestion. We agree that it might be a little bit disturbing to find H. pylori again in the cancer stem cell discussion. But, H. pylori being the main GC risk, it is involved in many “context/part” of gastric carcinogenesis. We did want to talk about CSCs in the H. pylori part since CSCs was not yet introduced in this part. Since this organisation seems to be good for the 3 other reviewers, we will keep the paragraph this way.

  1. In section 3.4 what are the other microorganisms involved in GC except H. Pylori? The authors have mentioned Clostridium, Fusobacterium and Lactobacillus but they aren’t elaborated properly. Rewrite the 3.4 section.

Thank you for your comments. In fact, no other bacteria apart from H. pylori is known to be involved in gastric carcinogenesis. To clarify this, we have reformulated the end of the paragraph as follows:

Line 594-596:

In GC patients, H. pylori is indeed more frequent and more abundant, corresponding to a cancer-specific signature, but, Clostridium, Fusobacterium and Lactobacillus species are also frequently abundant in those patients, though their role in gastric carcinogenesis is still not proven.”

Line 600-602:

These results are for the moment only descriptive and H. pylori still remains the only bacterium directly implicated in gastric carcinogenesis”

Indeed, there is no other bacteria except for H. pylori involved in gastric carcinogenesis. The gastric microbiota modifications described here are only descriptive and do not correspond to real implication in gastric carcinogenesis, but to modifications observed at the different carcinogenesis stages.

  1. Section 4.1: A table must show all the biomarkers and specific approaches to detect GC. Line 580-600: this stanza may be better suitable for microenvironment paragraphs. I don’t find HDGF as a biomarker in GC from the manuscript.

Thank you for your comments We have added a table for the biomarkers as well as for the CSC targeted strategies. HDGF is cited as biomarker line 666.

  1. First write about biomarkers then come to targeting approaches or therapy. Liquid biopsy biomarkers should come first then GCSC targeting.

Since overexpressing molecules can be potential biomarkers as well as targeting strategies, we had decided to discuss it this way and this is why we had combined this part. Since this is ok for all the 3 other reviewers, we will leave this part as such.

  1. Conclusion should be rewritten with specific biomarker and therapeutic approaches which are mostly used for GC at present time. Which approaches are better and should be considered in future for GC treatment?

We have added this sentence in the conclusion to talk about the only real targeted options that are used:

Line 1291-1294:

The complexity of this disease makes it such that despite research advances and all the highlighted potential biomarkers and therapeutical strategies, there are still only few targeted strategies like Trastuzumab and Ramucirumab used in clinic [98].”

Reviewer 3 Report

This review is summarizing gastric cancer, from histological and molecular classifications until new research and therapeutics. The article provides an excellent overview of the clinical features of GC, the connection with the risk factors and the gastric carcinogenesis as well as therapeutic strategies.

Strength: well written, timely, summarizing all existing literature.

The paper is an excellent conclusion of what is known so far in the literature.

This review article is exhaustive, and overall, very informative.

Limitations: summarizing the existing literature without giving overview tables and figures. Some references are needed in some paragraphs such as: line 30, 313, 498, 544, 554, 578, 744.

The authors give some information about the role of miRNAs in the GC that perhaps would be better under a different subtitle.

Author Response

Our point-by point response to the reviewer's comments are in bold under each comment.

Reviewer 3

This review is summarizing gastric cancer, from histological and molecular classifications until new research and therapeutics. The article provides an excellent overview of the clinical features of GC, the connection with the risk factors and the gastric carcinogenesis as well as therapeutic strategies.

Strength: well written, timely, summarizing all existing literature.

The paper is an excellent conclusion of what is known so far in the literature.

This review article is exhaustive, and overall, very informative.

Limitations: summarizing the existing literature without giving overview tables and figures. Some references are needed in some paragraphs such as: line 30, 313, 498, 544, 554, 578, 744.

Thank you very much for theses comments. In fact, we did figures summarising the main parts of the review but they were not integrated to the text, though submitted at the same time as the review.

We integrated these figures in the appropriate part of the text and cited them.

The following references were added as asked:

  • Line 30: [2] Cisto et al. & [3] Bosman et al.
  • Line 336 (ex-313): [55] Nguyen et al. 2017 & [56] Kodama et al. 2017
  • Line 560 (ex-498): [14] Corea et al 2012.
  • Line 616 (ex-544) : [1] Knight et al. 2019
  • Line 626 (ex-554) : [4] Guo et al. 2017, [8] Cristescu et al. 2015, [10] Li et al. 2017
  • Line 835 (ex-744): [98] Carrasco-Garcia et al. 2018, [118] Akin Telli et al. 2020

The authors give some information about the role of miRNAs in the GC that perhaps would be better under a different subtitle.

Thank you very much for you comment. In fact, we hesitated about putting miRNA in a different part. But since there is very little information on miRNA and that most involve CSCs we preferred putting it in the same chapter. We hope it’s ok for the reviewer.

Reviewer 4 Report

In the manuscript entitled “Gastric Cancer: advances in carcinogenesis research and new therapeutic strategies”, Seeneevassen et al. provides an extensive review regarding the impact of H. pylori, cancer stem cells, microenvironments, and microbiome on the gastric carcinogenesis and their potential clinical implications. The manuscript is well written and I have some comments.

The section of tumor microenvironments may be expanded by addressing recent single cell RNA-seq results that provide a comprehensive TME profiling and tumor cell subclassification. For example, Zhang et al. addressed the GC tumor cells can be subclassified into several subtypes, which are quite different from those from bulk-level RNA-seq (10.1136/gutjnl-2019-320368). Sathe et al. mentioned that classical M1/M2 models may not apply for macrophages in GC (10.1158/1078-0432.CCR-19-3231). In addition, I believe that the section should include the description of immune cells such as TIL, Tregs, DCs.

It is controversial whether the EBV+ GC tumors have good or poor prognosis (ref. 43), which requires explanation by the authors.

Author Response

Our point-by point response to the reviewer's comments are in bold under each comment.

Reviewer 4

In the manuscript entitled “Gastric Cancer: advances in carcinogenesis research and new therapeutic strategies”, Seeneevassen et al. provides an extensive review regarding the impact of H. pylori, cancer stem cells, microenvironments, and microbiome on the gastric carcinogenesis and their potential clinical implications. The manuscript is well written and I have some comments.

The section of tumor microenvironments may be expanded by addressing recent single cell RNA-seq results that provide a comprehensive TME profiling and tumor cell subclassification. For example, Zhang et al. addressed the GC tumor cells can be subclassified into several subtypes, which are quite different from those from bulk-level RNA-seq (10.1136/gutjnl-2019-320368). Sathe et al. mentioned that classical M1/M2 models may not apply for macrophages in GC (10.1158/1078-0432.CCR-19-3231). In addition, I believe that the section should include the description of immune cells such as TIL, Tregs, DCs.

Thank you very much for your comment and suggestions. We decided to add the Zhang et al. reference to the classification part as follows since it talks much about classification and we think it fits more there:

(Line 79-92):

“Finally, recent developments in single cell methods allows more precise deciphering of GC cells heterogeneity, though not applicable as routine diagnosis. Zhang et al., propose a GC transcriptome atlas after analysis of 22,677 cells from 3 non-tumorous samples and 9 tumours [11]. Comparison of malignant epithelium to no malignant ones put into evidence overexpression of tumour specific genes like REG4, TFF3, CLDN4 and CLDN7. Genes involved in gastric mucus and enzyme secretion like GKN1, PGC, MUC5AC and LIPF were more expressed in non-tumorous epithelium [11]. In order to compare the cellular genetic profiles obtained to known histopathological classifications, tumour samples with known histology were added to analysis. Analysed malignant cells clustered 5 main cell groups: C1 having 96.9% cells from diffuse-type sample, C2 composed of 97.1% cells from intestinal-type sample, C3 with mixed-type and intestinal- type cells, C4 containing cells mainly from one of the intestinal-type samples and C5 with cells from EBV+ patients. Interestingly, C4/C5 seem to be correspond to novel subtypes, molecularly different from C1, C2 and C3 which correspond to the Lauren classification subtypes [11].”

We then added the following paragraphs to the tumour microenvironment section to expand it, as asked:

Lines 433-439:

Most immune cells infiltrating tumours are Tumour Infiltrating Lymphocytes (TILs) consisting of CD3+ Tcells (CD8+ Tcytotoxic and CD4+ Thelper cells) as well as FOXP3+ Treg cells. The number of TILs present in tumours reflects the host immune response mechanism. Indeed, Yu et al. demonstrated that high CD8+ and CD3+ T cells infiltration in tumours are associated to better prognosis of patients compared to low infiltration [68]. Interestingly, CD8+ T cells seem to be less present in diffuse/mixed types GC compared to intestinal type and might be associated to worse outcome [69].”

Lines 493-500:

“Interestingly, recent single cell tumour microenvironment genomic characterisation demonstrate that GC TAMs genetic signatures do not match those of known M1 or M2 macrophages [82]. M1 and M2 known distinguishing genetic markers are co-expressed in the same macrophage cluster. Nevertheless, genes distinguishing these different TAM clusters seem to be related to the HSP family, THBS1, MMPs family, CCL20, CCL18 and CCL3 chemokines among others, as well as cell cycle dependent genes. In addition, these TAMs seem to differ from PBMC monocytes but conserve some similarities with normal tissue macrophages [82].”

Lines 514-517:

“In relation with immunosuppression mechanisms, Regulatory T cells (Tregs) seem to be abundant in gastric TME [82,85]. Sathe et al. single cell analysis of gastric TME puts into light two different Treg classes marked by different expression level of proliferation-associated genes.”

Lines 528-525:

“Dendritic cells (DCs) can also reflect GC patient prognosis. Liu et al., showed that different subsets of DCs, plasmacytoid DCs (pDCs) and myeloid CD1+ DCs (mDC1s) are increased in GC patients peripheral blood compared to healthy controls [86]. Also, high level of pDCs seems to correlate with GC stage progression while mDC1 level does not change. Furthermore, different DC subtypes are found according to GC stages. Early GC T stages present high peripheral DC2 number and tumour infiltrating DC1/DC2 ratio while low level of tumour infiltrating DC2s and high DC1/DC2 ratio are observed at N0 GC stage [87].”

It is controversial whether the EBV+ GC tumors have good or poor prognosis (ref. 43), which requires explanation by the authors.

Yes, in fact EBV+ GC tumours are associated to a better prognosis when compared to other GC subtypes, especially diffuse/mixed histological subtypes. We added the reference [16 - Huang et al.] and changed the sentence (line 115) to “EBV-positive GC are mostly associated to MSI molecular type and are related to less worst survival prognosis than diffuse/mixed GC subtypes” to better support this statement.

But at the same time, according to reference [45 (ex-reference 43)] high EBV load in tumours is correlated to high PDL1 expression and bad prognosis of patients when compared to low EBV load. This indeed might seem controversial but in fact it means that among patients with EBV+ tumours, those having high viral load survive less than those having lower viral load.

According to Huang et al. EBV+ phenotype is not different from other phenotypes only in the fact that the tumours are EBV+ but these tumours also present clinicopathological differences such as “age, gender, stump cancer, gastric location, tumour size and HER2 status” amongst others, which might explain the positive prognosis compared to other phenotypes though high viral load is of negative prognosis.

To clarify this in the text, we modified the sentence (line 285-288) to:

“A recent study showed that among EBV-positive tumours, those with high viral load was correlated to higher tumour cells PD-L1 expression and worse patient prognosis compared to those with lower EBV viral load”